# Experimental Investigation on the Quasi-Static Tensile Capacity of Engineered Cementitious Composites Reinforced with Steel Grid and Fibers

**DOI:** 10.3390/ma12172666

**Published:** 2019-08-21

**Authors:** Liang Li, Wenli Liu, Jun Wu, Wenjie Wu, Meng Wu

**Affiliations:** 1Key Laboratory of Urban Security and Disaster Engineering, Beijing University of Technology, Ministry of Education, Beijing 100124, China; 2School of Urban Railway Transportation, Shanghai University of Engineering Science, Shanghai 201620, China

**Keywords:** engineered cementitious composites, steel grid, fiber, tensile capacity, energy dissipation

## Abstract

An engineered cementitious composite (ECC) was reinforced with a steel grid and fibers to improve its tensile strength and ductility. A series of tensile tests have been carried out to investigate the quasi-static tensile capacity of the reinforced ECC. The quasi-static tensile capacities of reinforced ECCs with different numbers of steel-grid layers, types of fibers (Polyvinyl alcohol (PVA) fiber, KEVLAR fiber, and polyethylene (PE) fiber), and volume fractions of fibers have been tested and compared. It is indicated by the test results that: (1) On the whole, the steel grid-PVA fiber and steel grid-KEVLAR fiber reinforced ECCs have high tensile strength and considerable energy dissipation performance, while the steel grid-PE fiber reinforced ECC exhibits excellent ductility. (2) The ultimate tensile strength of the reinforced ECC can be improved by the addition of steel grids. The maximal peak tensile stress increase is about 50–95% or 140–190% by adding one layer or two layers of steel grid, respectively. (3) The ultimate tensile strength of the reinforced ECC can be enhanced with the increase of fiber volume fraction. For a certain kind of fiber, a volume fraction between 1.5% and 2% grants the reinforced ECC the best tensile strength. Near the ultimate loading point, the reinforced ECC exhibits strain hardening behavior, and its peak tensile stress increases considerably. The energy dissipation performance of the reinforced ECC can also be remarkably enhanced by such an increase in fiber volume fraction. (4) The ductility of the steel grid-PVA fiber reinforced ECC can be improved by the addition of steel grids and the increase of fiber volume fraction. The ductility of the steel grid-KEVLAR fiber reinforced ECC can be improved by the addition of steel grids alone. The ductility and energy dissipation performance of the steel grid-PE fiber reinforced ECC can be improved with the increase of fiber volume fraction alone. A mechanical model for the quasi-static initial and ultimate tensile strength of the steel grid-fiber reinforced ECC is proposed. The model is validated by the test data from the quasi-static tension experiments on the steel grid-PE fiber reinforced ECC.

## 1. Introduction

Engineered cementitious composites (ECCs) have been used widely in the civil engineering and transportation applications, such as airport runways. They have a higher tensile strength and ductility compared to normal concrete. They also exhibit high energy dissipation performance due to strain hardening and multiple-cracking behavior [1]. During the past decades, various types of ECCs with different ingredients have been developed. The mechanical properties of these ECCs have also been the subject of intense research during the past decades.

Various types of fibers, such as carbon fibers, steel fibers, and polymer fibers have been added to the ECCs to improve their tensile capacity and energy dissipation performance. Hybrid fibers, namely, the combination of different types of fibers, have also been adopted to gain the composite effect. Tran and Kim [2] investigated the direct tensile stress versus strain response of high-performance fiber-reinforced cementitious composites (HPFRCCs) at high strain rates between 10s^−1^ and 40s^−1^. Twisted and hooked steel fibers were used in the HPFRCCs. Arboleda et al. [3] studied the tensile behavior of fabric-reinforced cementitious matrix composites. Different fabrics, including polyparaphenylene benzobisoxazole (PBO), carbon, glass, and carbon and glass with a special protective coating were used for the investigation. Kim et al. [4] conducted direct tensile and shear transfer tests of amorphous micro steel (AMS) fiber-reinforced cementitious composites. Ali et al. [5,6] investigated the behavior under impact loading of a hybrid fiber-reinforced ECC incorporating short, randomly dispersed shape memory alloy (HECC-SMAF) and PVA fibers by the drop weight impact test and numerical simulation. Yu et al. [7,8] developed ultra-high ductile cementitious composites (UHDCCs) with the polyethylene (PE) fibers. It was reported that the tested UHDCCs exhibited an average tensile strain of 8% at peak stress. The rate sensitivity of the UHDCCs was evaluated by direct tensile experiments under different strain rates. Curosu et al. [9] investigated the tensile behavior of high-strength strain-hardening cement-based composites (HS-SHCCs) made with four different types of dispersed high-performance polymer fibers.

Zhou et al. [10] investigated the mechanical properties of hybrid ECCs incorporating steel and polyethylene fibers. Zhang et al. [11] studied the mechanical properties and carbonation durability of ECCs reinforced by polypropylene and hydrophilic polyvinyl alcohol fibers. Kim et al. [12] investigated the hybrid effect of twisted steel and polyethylene fibers on the tensile performance of ECCs. Zhu et al. [13] conducted uniaxial tensile tests to investigate the stress-strain behavior of carbon-fiber grid-reinforced ECCs. Al-Gemeel et al. [14] and Li et al. [15] investigated the tensile behavior of a basalt textile grid reinforced ECC. Sun et al. [16] conducted a series of tests to study the mechanical behavior of the ECCs reinforced with polyvinyl alcohol (PVA) fibers.

Some research has been focused on the factors that impact the tensile performance of ECCs. Wang et al. [17] studied the tensile performance of polyvinyl alcohol (PVA)-steel hybrid fiber reinforced ECCs, focusing on the impacts of steel-fiber content and water-to-binder ratio of the matrix. Abrishambaf et al. [18] investigated the influence of fiber orientation on the tensile behavior of ultra-high-performance fiber-reinforced cementitious composites. Wu et al. [19] studied the effect of the morphological parameters of natural sand on the mechanical properties of ECCs. The mechanical behavior of ECC and fiber-reinforced ECC at high temperatures has also been investigated for solar emission, fire, gas explosion, and blast scenarios [20,21].

Even the tensile strength and ductility of normal ECC or fiber-reinforced ECC are not sufficient to resist strong blast and impact loads. The tensile strength and energy dissipation performance of ECC can be improved by adding a steel grid and fibers to the matrix. In the current study, the ECC is reinforced with a steel grid and fibers to improve its tensile strength and ductility. A series of experiments are carried out to investigate the quasi-static tensile strength of the reinforced ECC using a Z100 material tensile testing machine manufactured by the Zwick/Roell Group (Ulm, Germany). The quasi-static tensile strength and energy dissipation performance of the various reinforced ECCs are tested and compared. The test variations include the number of steel-grid layers (one layer, two layers), the type of fibers (polyvinyl alcohol (PVA) fiber, KEVLAR fiber, and polyethylene (PE) fiber), and the volume fraction of fibers (0%, 0.5%, 1%, 1.5%, 2%). A mechanical model for the quasi-static initial and ultimate tensile strength of the steel grid-fiber reinforced ECC is proposed. This model is validated by the test data from the quasi-static tension experiments on the steel grid-PE fiber reinforced ECC.

## 2. Experimental Program

### 2.1. Materials and Mixture Proportions

The raw materials used in the current study for the matrix of the ECC include P.O42.5 ordinary Portland cement, siliceous fly ash, water, and superplasticizer. Three types of fibers, (polyvinyl alcohol (PVA) fibers, KEVLAR fibers, and polyethylene (PE) fibers) and two different steel-grid configurations (one layer and two layer) were added to the matrix to improve its tensile strength and ductility. The physical and mechanical properties of P.O42.5 ordinary Portland cement are listed in Table 1. The term “initial setting time” refers to the time required for the cement slurry to begin losing plasticity, and the term “final setting time” refers to the time required for the cement slurry to completely lose its plasticity and to begin to exhibit considerable strength. The three types of fibers used in the experiments are shown in Figure 1, and their basic physical parameters are presented in Table 2, Table 3 and Table 4. The steel grids used in the experiments are shown in Figure 2, the one-layer steel grid is on the right side, and the two-layer one is on the left side. The diameter of steel grid wire is 0.88 mm, the dimensions of the grid holes are 12.83 mm × 12.76 mm.

The mass mixture proportions of cement, silica ash, water, and superplasticizer for the matrix of the ECC are listed in Table 5. The matrix is reinforced with one layer or two layers of steel grid, respectively. In addition, one of the three types of fibers mentioned above is added to the matrix of the ECC. The volume contents of fiber in the current experimental study are 0%, 0.5%, 1%, 1.5%, and 2.0%, respectively.

### 2.2. Specimen Fabrication and Test Setup

A rectangular, thin-plate-shaped specimen is adopted in the current experimental study, and the dimensions of the specimen are 300 mm × 75 mm × 20 mm. A steel mold is used for the casting of the specimens. The specimen fabrication procedure can be described as follows:(1).Spread lubricant on the inner surface of the mold for the convenience of the demolding process. Install the steel grid in the middle part of the mold.(2).Weigh-up the material ingredients of the tested ECCs, including cement, siliceous fly ash, water, superplasticizer, and fibers according to the mixture proportion. When the KEVLAR fiber is used, a cleaning process should be carried out with alcohol to remove grease from the fiber surfaces.(3).The cement and siliceous fly ash are first dry-mixed in the mixer for 3 min. The superplasticizer is mixed with the water, and they are then added into the dry mixture and mixed for a further 2 min to produce a consistent and uniform matrix. The fibers are then added into the matrix and mixed for an additional 3 min to make the fibers spread in the mixture until they reach a uniform state.(4).The fresh ECC mixtures are cast into the steel molds. The molds are then placed on a shake-table to eject entrapped air and produce a denser matrix by the vibration of the table-board. The specimens are demolded after 24 h and then set in a curing room for 28 days, where the temperature is 20 ± 0.5 °C, and the relative humidity is 95 ± 5%.(5).When the curing process is finished, thin steel pieces are adhered to the ends of the specimens with epoxy resin adhesive in order to provide extra reinforcement at the location where the specimens are connected to the tensile-loading machine. The specimens with the end reinforcement are shown in Figure 3.

The Z100 universal material testing machine manufactured by the Zwick/Roell Group (Ulm, Germany) is used for the tensile tests of the ECCs reinforced with steel grid and fibers. The displacement-controlled loading mode is adopted for the test. The tensile load is measured by the 100kN force transducer. A pair of automatic extensometers is used to measure the strain of the specimen. The measurement range of each extensometer is 100 mm.

The tensile loading rate is 0.1mm/min, and the corresponding strain rate is 1 × 10^−5^ s^−1^. The tensile stress and strain of the specimen can be computed as Equations (1) and (2):(1)σ=F/bh
(2)$$\varepsilon =L/L_{0}$$
where *σ* and *ε* are the tensile stress and strain of the specimen, *F* is the measured tensile force, *b* and *h* are the width and thickness of the specimen, respectively. In the current test study, *b* = 75 mm, *h* = 20 mm. *L* is the measured tensile deformation value of the specimen, and *L_0_* is the standard distance between two extensometers, in the current test *L_0_* = 100 mm.

## 3. Test Results and Discussion

### 3.1. Quasi-Static Tensile Test Results of the Steel Grid-PVAFiber Reinforced ECC

The quasi-static tensile test results of the ECC reinforced with steel grids and PVA fibers are presented in Table 6. Because ECC exhibits considerable strength and ductility even after reaching its peak tensile stress, during loading, the failure of ECC is considered to occur when the tensile stress has descended to 80% of its peak value. The strain corresponding to the 80% of the peak stress in the descending segment of the stress-strain curve is defined as ultimate strain. The energy dissipation can be computed from the area enveloped by the stress–strain curve. In the specimen notation, “M” stands for the matrix, “S” stands for the steel grid, “A” stands for the PVA fiber, and the two numbers stand for the volume content of the PVA fibers and the number of steel-grid layers, respectively. For example, “A0.5S2” stands for the ECC specimen with a volume content of the PVA fibers of 0.5% and two layers of steel grid.

A comparison of the energy dissipation of different types of steel grid-PVA fiber reinforced specimens is shown in Figure 4. It is illustrated that the energy dissipation performance of the ECC can be improved remarkably by the addition of a steel grid. For the ECC matrix specimens, only a single, main crack is observed and the specimen splits along the crack when the failure occurs. For the steel grid-PVA fiber reinforced ECC specimens, however, a multiple-cracking phenomenon can be observed during the tensile loading. Some post-failure specimens with multiple cracks are shown in Figure 5. After the appearance of the first crack, the tension stress in the cracking region is mainly borne by the steel grid, which restrains the further development of a single, critical crack.

A comparison of the stress-strain curves of the specimens with different numbers of steel-grid layers is shown in Figure 6. A comparison for the specimens with different fiber volume fractions is shown in Figure 7. These results indicate that the tensile strength of the PVA-ECC can be enhanced by the addition of steel grids. For the fiber volume fraction of 1.5%, the maximal peak tensile stress increase of about 95% or 160% compared to the matrix specimen by adding one layer or two layers of steel grid can be obtained, respectively. In addition, the ductility of the PVA-ECC can also be improved to some extent by the addition of steel grids. The ultimate strain increases by about 0.6% or 0.8% by adding one layer or two layers of steel grid, respectively. This means that the ultimate strain of the steel-grid specimens is increased by 0.6% or 0.8% absolute strain compared to the matrix specimens. After the peak stress, the steel grids begin to separate from the matrix, and this leads to a deformation inconsistency between the steel grids and the matrix. The peak tensile stress of the steel grid-PVA fiber reinforced ECC can be enhanced with the increase of the volume fraction of PVA fibers. For the volume fractions of 0.5%, 1%, 1.5%, and 2%, the peak tensile stress increases by about 45%, 80%, 85%, and 170% compared to the matrix specimen, respectively. The ductility can also be improved with the increase of the volume fraction of PVA fibers.

### 3.2. Quasi-Static Tensile Test Results of the Steel Grid-KEVLARFiber Reinforced ECC

The quasi-static tensile test results of the ECC reinforced with steel grids and KEVLAR fibers are presented in Table 7. In the specimen notation, “M” stands for the matrix, “S” stands for the steel grid, “K” stands for the KEVLAR fibers, and the two numbers stand for the volume content of the KEVLAR fibers and the number of steel-grid layers, respectively. For example, “K1S2” stands for the ECC specimen with a volume content of the KEVLAR fibers of 1% and two layers of steel grid.

The comparison of the energy dissipation of different types of steel grid-KEVLAR fiber reinforced specimens is shown in Figure 8. It is illustrated that the energy dissipation performance of the ECC can be improved remarkably by the addition of a steel grid. When two layers of steel grid are added to the matrix, the energy dissipation performance increases by about two to eight times.

A comparison of the stress-strain curves of the specimens with different numbers of steel-grid layers is shown in Figure 9. A comparison for the specimens with different fiber volume fractions is shown in Figure 10. The results indicate that the tensile strength of the KEVLAR-ECC can be enhanced by the addition of steel grids. For the fiber volume fraction of 1.0%, the maximal peak tensile stress increase of about 50% or 140% compared to the matrix specimen by adding one layer or two layers of steel grid can be obtained, respectively. In addition, the ductility of the KEVLAR-ECC can also be improved noticeably by the addition of steel grids. For the KEVLAR-ECC specimens, the ultimate strain increases by about two or 3.5 times compared to the matrix specimen by adding one layer or two layers of steel grid, respectively. The peak tensile stress of the steel grid-KEVLAR fiber reinforced ECC can be enhanced with the increase of the volume fraction of KEVLAR fibers. For the volume fractions of 0.5%, 1%, 1.5%, and 2%, the peak tensile stress increases by about 80%, 65%, 125%, and 200% compared to the matrix specimen, respectively. The tensile strength of the KEVLAR-ECC has a maximum average increase when the volume fraction of the KEVLAR fibers is 2%. But relative to the PVA-ECC, the ductility of the KEVLAR-ECC is poor for the volume fractions of 1.5% and 2%. For instance, the ultimate strain of K1.5S2 is 70% of that of A1.5S2, the ultimate strain of K2S1 is about 52% of that of A2S1. This may be because the surface of KEVLAR fiber has not been adequately cleaned, and this leads to a relatively weak bond between the KEVLAR fiber and matrix.

### 3.3. Quasi-Static Tensile Test Results of the Steel Grid-PEFiber Reinforced ECC

The quasi-static tensile test results of the ECC reinforced with steel grids and PE fibers are presented in Table 8. In the specimen notation, “M” stands for the matrix, “S” stands for the steel grid, “E” stands for the PE fiber, and the two numbers stand for the volume content of the PE fibers and the number of steel-grid layers, respectively. For example, “E2S2” stands for the ECC specimen with a volume content of the PE fibers of 2% and two layers of steel grid.

A comparison of the energy dissipation of different types of steel grid-PE fiber reinforced specimens is shown in Figure 11. It is shown that the PE-ECC with a fiber volume fraction of 1–2% exhibits excellent energy dissipation performance. A comparison of the stress-strain curves of the specimens with different numbers of steel-grid layers is shown in Figure 12. A comparison for the specimens with different fiber volume fractions is shown in Figure 13. The results indicate that the tensile strength of the PE-ECC can be enhanced by the addition of steel grids. For the fiber volume fraction of 0.5%, the maximal peak tensile stress increase of about 80% or 190% compared to the matrix specimen by adding one layer or two layers of steel grid can be obtained, respectively. On the other hand, the ductility of the PE-ECC has not been improved remarkably by the addition of steel grids. For the specimens with a fiber volume fraction of 1–2%, the ultimate strain decreases when the steel grid is added to the matrix. At the initial stage of tensile loading, the steel grid has a strong bond with the matrix. After the peak stress, the steel grid will separate from the matrix, and this leads to the deformation inconsistency between the steel grid and the matrix. The results indicate that the ultimate strain mainly depends on the deformation capacity of the steel grid. The deformation capacity of the steel grid is, however, lower than that of the PE-ECC with a relatively high fiber volume fraction, such as 1.5% or 2%. The PE-ECC without steel-grid reinforcement shows a ductile failure behavior. The ductility and energy dissipation performance can be improved with the increase of the volume fraction of PE fibers. The peak tensile stress of the steel grid-PE fiber reinforced ECC can be enhanced with the increase of the volume fraction of PE fibers. For the volume fraction of 0.5%, 1%, 1.5%, and 2%, the peak tensile stress increases by about 70%, 80%, 130%, and 160% compared to the matrix specimen, respectively.

## 4. Mechanical Model for Quasi-Static Tensile Strength of Steel Grid-Fiber Reinforced ECC

In this section, a mechanical model for the quasi-static initial and ultimate tensile strength of the steel grid-fiber reinforced ECC is proposed. The initial tensile strength corresponds to the tensile stress when the first crack appears in the specimen. The steel grid-fiber reinforced ECC is treated as a composite material consisting of steel grid and ECC matrix. The total tensile strength of the steel grid-fiber reinforced ECC can be acquired from the sum of the tensile strength of steel grid and that of ECC matrix.

### 4.1. Model Development

The stress-strain relation of the steel grid for the tensile loading can be treated as an ideal hyperbolic model as shown in Figure 14. It can be expressed as Equations (3) and (4):(3)$$\sigma_{s}=\varepsilon E_{s}\;\varepsilon <\varepsilon_{y}.$$
(4)$$\sigma_{s}=\sigma_{y}+\left(\varepsilon -\varepsilon_{y}\right)E_{t}\;\varepsilon_{y}\leq \varepsilon \leq \varepsilon_{u}.$$
where *σ_s_* and *ε* are the tensile stress and strain of the steel grid, respectively. *σ_y_* is the yield stress of the steel grid, *ε_y_* and *ε**_u_* are the yield strain and ultimate strain of the steel grid, respectively. *E_s_* and *E_t_* are the elastic modulus of the steel grid at the pre-yield and post-yield stage, respectively.

The tensile strength of ECC matrix consists of the strength related to the crack end toughness and that related to the fiber bonding. From the fracture mechanics theory, the stress strength factor of a straight crack under the action of tensile loads can be expressed as Equation (5) [22]:
(5)$$K_{L}=\sigma_{L}\sqrt{2w\tan\left(\frac{\pi c}{2w}\right)}$$
where *w* is the width of the specimen, *c* is the length of the crack, and *σ_L_* is the tensile stress acting on the ECC matrix. For a composite material, the stress strength factor *K_L_* equals the crack end toughness of the composite material *K_tip_*. The tensile stress *σ_L_* can, thus, be written as Equation (6):(6)$$\sigma_{L}=\frac{K_{tip}}{\sqrt{2w\tan\left(\frac{\pi c}{2w}\right)}}$$

The crack end toughness of the composite material *K_tip_* can be expressed as Equation (7) [23]:(7)$$K_{tip}=\frac{E_{c}}{E_{m}}K_{m}$$
where *E_c_* and *E_m_* are the elastic modulus of the composite material and matrix, respectively. *K_m_* is the crack end toughness of the matrix.

For a single fiber in the ECC matrix, the relation between bonding load *P_f_* and crack split displacement δ during the debonding process can be expressed as Equations (8) and (9) [24]:(8)$$P_{f}\left(\delta \right)=\frac{\pi }{2}\sqrt{\left(1+\eta \right)E_{f}d_{f}^{3}\tau \delta }e^{\phi }\;\delta \leq \delta_{0}$$
(9)$$P_{f}\left(\delta \right)=\pi \tau ld_{f}\left(1-\frac{\delta }{l}\right)e^{\phi }\;\delta_{0}<\delta \leq l$$
where $\delta_{0}=4l^{2}\tau /(1+\eta )E_{f}d_{f}$ is the crack split displacement of the fiber with a length of *l* when full debonding occurs. *d_f_* is the diameter of the fiber, *τ* is the bonding strength of the fiber-matrix interface. $\eta =V_{f}E_{f}/V_{m}E_{m}$, where *V_f_* and *V_m_* are the volume fractions of fiber and matrix, respectively, and *E_f_* is the elastic modulus of the fiber. *ϕ* is the angle between fiber orientation and the acting direction of the bonding load *P**_f_*.

The total fiber bonding stress of the fiber-reinforced ECC matrix can be obtained by the integration of bonding load *P**_f_*. It can be written as Equation (10) [25]:(10)$$\sigma_{B}\left(\delta \right)=\frac{4V_{f}}{\pi d_{f}^{2}}\int_{\phi =0}^{\pi /2}\int_{z=0}^{\left(l\cos\phi \right)/2}P_{f}\left(\delta \right)p\left(\phi \right)p\left(z\right)dzd\phi$$
where *p(ϕ)* and *p(z)* are the probability density functions of fiber orientation angle *ϕ* and fiber centroid position *z*, respectively. For three-dimensional randomly distributed fibers, these two probability density functions are expressed as Equations (11) and (12):(11)$$p(z)=\frac{2}{l}\;0\leq z\leq \frac{l}{2}\cos\phi$$
(12)$$p(\phi )=\sin\phi \;0\leq \phi \leq \frac{\pi }{2}$$

Substituting Equations (8), (11), and (12) into Equation (10) results in the expression of the total fiber bonding stress of fiber-reinforced ECC before the full debonding as follows Equation (13):(13)$$\sigma_{B}\left(\delta \right)=\sigma_{0}g\left[2{\left(\frac{\delta }{\delta *}\right)}^{1/2}-\frac{\delta }{\delta *}\right]\;\delta \leq \delta *$$
where $\delta *=l^{2}\tau /(1+\eta )E_{f}d_{f}$ is the crack split displacement of the fiber with a length of *l*/2 when full debonding occurs, $\sigma_{0}=\tau V_{f}l/2d_{f}$, and *g* is the buffer factor and can be expressed as Equation (14):(14)$$g=\frac{2}{4+f^{2}}\left(1+e^{\pi f/2}\right)$$
where *f* is the buffer coefficient. In the current study, *f* = 0.8 [26].

The crack split displacement δ can be expressed as Equation (15) [23]:(15)$$\delta =\delta_{a}\sqrt{c\left(1-\frac{r^{2}}{c^{2}}\right)}$$
where $\delta_{a}=2K_{m}(1-\nu^{2})/E_{m}\sqrt{\pi }$, and *v* is the Poisson ratio of the ECC matrix.

Substituting Equation (15) into Equation (13) results in the bonding stress of a single fiber, and the total fiber bonding force on the crack can be expressed as Equation (16):(16)$$F_{B}=2\int_{0}^{c}\sigma_{B}(\delta )dr$$

*F_B_* corresponds to a specimen width of 2*w*. The integration of Equation (16) results in the expressions of the tensile stress related to fiber bonding as follows Equations (17) and (18):(17)σF=cwσ0g[1.748×δaδ*(c)14−π4δaδ*c]    c≤w
(18)σF=σ0g[1.748×δaδ*(c)14−π4δaδ*c]    c>w

Taking both the crack end toughness and fiber bonding into consideration results in the tensile stress-crack split displacement relation as follows Equations (19) and (20):(19)σc=Ktip2wtan(πc2w)+cwσ0g[1.748×δaδ*(c)14−π4δaδ*c]    c<w
(20)σc=σ0g[1.748×δaδ*(c)14−π4δaδ*c]    c≥w

When the length of the crack *c* is greater than the width of the specimen *w*, the crack end toughness is not taken into consideration, and the *K_tip_* term is deleted from the tensile stress-crack split displacement relation. Letting *c = w*, we can obtain the initial tensile strength of the ECC matrix as follows Equation (21):(21)σfc=Ktip2w+σ0g[1.748×δaδ*(w)14−π4δaδ*w]

For the tensile ultimate state of the ECC matrix, the crack end toughness is neglected. Letting *δ* = *δ^*^*, we can obtain the ultimate tensile strength of the ECC matrix as follows Equation (22):(22)$$\sigma_{tu}=0.963\times \sigma_{0}g$$

The initial tensile strength of steel grid-fiber reinforced ECC can be obtained by the sum of the initial tensile strength of the steel grid and that of the ECC matrix. It can be expressed as follows Equation (23):(23)σfc=FScAc+Ktip2w+σ0g[1.748×δaδ*(c)14−π4δaδ*c]
where *F_Sc_* is the tensile force carried by steel grid when the first crack appears in the specimen, and *A_c_* is the cross-sectional area of specimen.

The ultimate tensile strength of steel grid-fiber reinforced ECC can be expressed as follows Equation (24):(24)$$\sigma_{tu}=\frac{F_{Su}}{A_{c}}+0.963\times \sigma_{0}g$$
where *F_Su_* is the ultimate tensile force of the steel grid.

### 4.2. Model Validation

The mechanical model for the quasi-static initial and ultimate tensile strength of the steel grid-fiber reinforced ECC proposed in the current study is validated by the quasi-static tension test data of the steel grid-PE fiber reinforced ECC. The comparisons of the analytical and experimental initial and ultimate tensile strengths of the steel grid-PE fiber reinforced ECC are presented in Table 9 and Table 10, respectively.

It is shown that, as a whole, the analytical initial tensile strength results predicted by the proposed mechanical model agree well with the corresponding experimental results, with most error below 20%. The analytical ultimate tensile strength results predicted by the proposed mechanical model are, however, larger than the corresponding experimental results. This means that the proposed mechanical model overestimates the ultimate tensile strength of the steel grid-fiber reinforced ECC. This is because the slip and debonding between the steel grid and the ECC matrix during the tensile loading are not taken into consideration in the proposed mechanical model. The tensile stress of steel grid increases after the peak stress in the proposed model, which is contrary to the observed test behavior of the steel grid. In the tensile tests, the tensile stress of the steel grid decreases after the peak stress.

## 5. Conclusions

In this research, an engineered cementitious composite (ECC) was reinforced with steel grids and different types of fibers to improve its tensile strength and ductility. A series of tensile tests have been carried out to investigate the quasi-static tensile capacity of the reinforced ECC using a Z100 material tensile testing machine manufactured by the Zwick/Roell Group of Germany. The quasi-static tensile capacity of reinforced ECCs with different numbers of steel-grid layers, types of fibers (PVA fibers, KEVLAR fibers, and PE fibers), and volume fractions of fibers have been tested and compared. It is indicated by the test results that:(1).On the whole, the steel grid-PVA fiber, steel grid-KEVLAR fiber, and steel grid-PE fiber reinforced ECCs all have higher tensile strength and ductility than the ECC matrix. They can also exhibit excellent energy dissipation performance.(2).The ultimate tensile strength of the reinforced ECC can be improved by the addition of steel grids. For the steel grid-PVA fiber reinforced ECC, when the fiber volume fraction is 1.5%, a maximal peak tensile stress increase of about 95% or 160% compared to the matrix specimen by adding one layer or two layers of steel grid can be obtained, respectively. For the steel grid-KEVLAR fiber reinforced ECC, when the fiber volume fraction is 1.0%, the maximal peak tensile stress increase of about 50% or 140% compared to the matrix specimen by adding one layer or two layers of steel grid can be obtained, respectively. For the steel grid-PE fiber reinforced ECC, when the fiber volume fraction is 0.5%, the maximal peak tensile stress increase of about 80% or 190% compared to the matrix specimen by adding one layer or two layers of steel grid can be obtained, respectively. A more remarkable increase of ultimate tensile strength can be obtained by adding two layers of steel grid, but it is more difficult to have a firm bonding with the ECC matrix for two layers of steel grid than one layer of steel grid.(3).The ultimate tensile strength of the reinforced ECC can be enhanced with the increase of fiber volume fraction. For all of the fiber types investigated, a volume fraction between 1.5% and 2% can make the reinforced ECC gain the best tensile strength. With these higher fiber volume fractions, the reinforced ECC exhibits strain hardening behavior, and its peak tensile stress increases considerably. The energy dissipation performance of the reinforced ECC can also be enhanced remarkably.(4).The ductility of PVA fiber reinforced ECC can be improved by the addition of steel grids and the increase of fiber volume fraction. The phenomenon of multiple-cracking can be observed for steel grid-PVA fiber reinforced ECC. The steel grid-PE fiber reinforced ECC also exhibits significant ductility and energy dissipation performance. Under those circumstances when excellent ductility and energy dissipation performance are required, it is better to use PVA fibers or PE fibers. The ductility of the steel grid-KEVLAR fiber reinforced ECC can be improved by the addition of steel grids. The ductility and energy dissipation performance of the steel grid-PE fiber reinforced ECC can be improved with the increase of fiber volume fraction.

A mechanical model for the quasi-static initial and ultimate tensile strengths of the steel grid-fiber reinforced ECC is proposed. This model is validated by the quasi-static tension test data of the steel grid-PE fiber reinforced ECC. It is indicated by the comparison of the analytical tensile strengths with the corresponding test results that the initial tensile strength predicted by the proposed mechanical model is relatively accurate, but the analytical ultimate tensile strength results predicted by the proposed mechanical model are larger than the corresponding experimental results.

## Figures and Tables

**Figure 1 materials-12-02666-f001:**
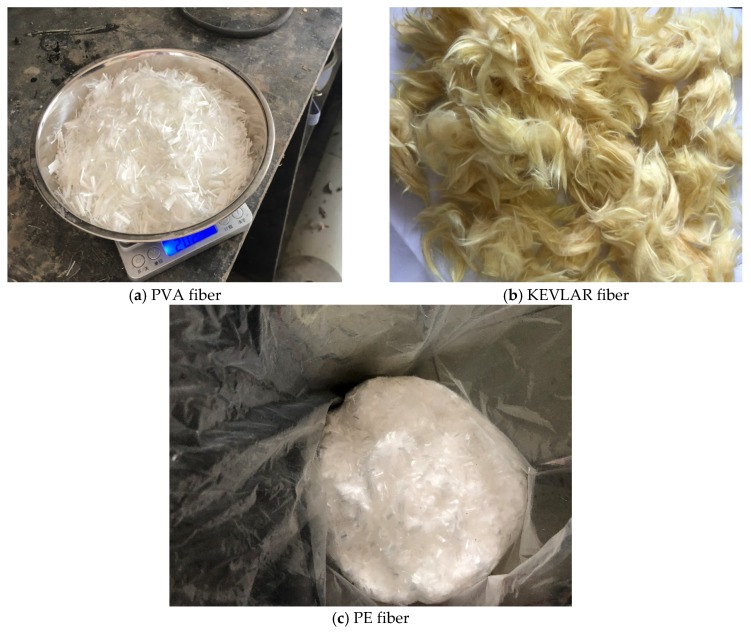
Three types of fibers used in the current study. (**a**) PVA fiber; (**b**) KEVLAR fiber; (**c**) PE fiber.

**Figure 2 materials-12-02666-f002:**
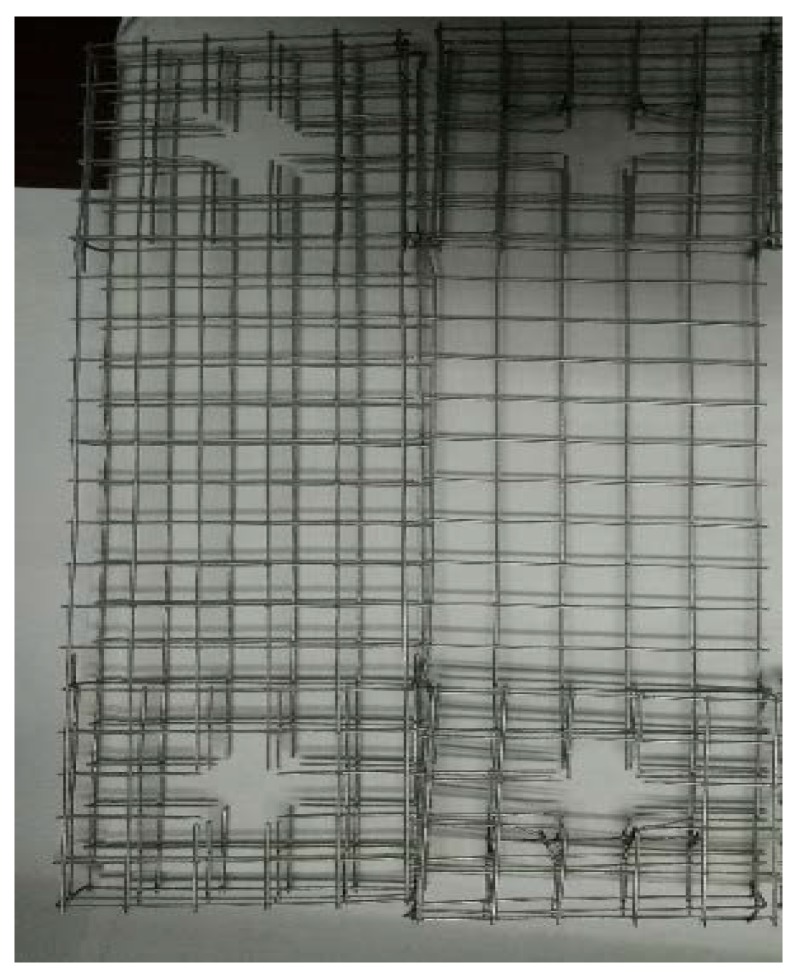
Double-layer (**left**) and single-layer (**right**) steel grids used in the current study.

**Figure 3 materials-12-02666-f003:**
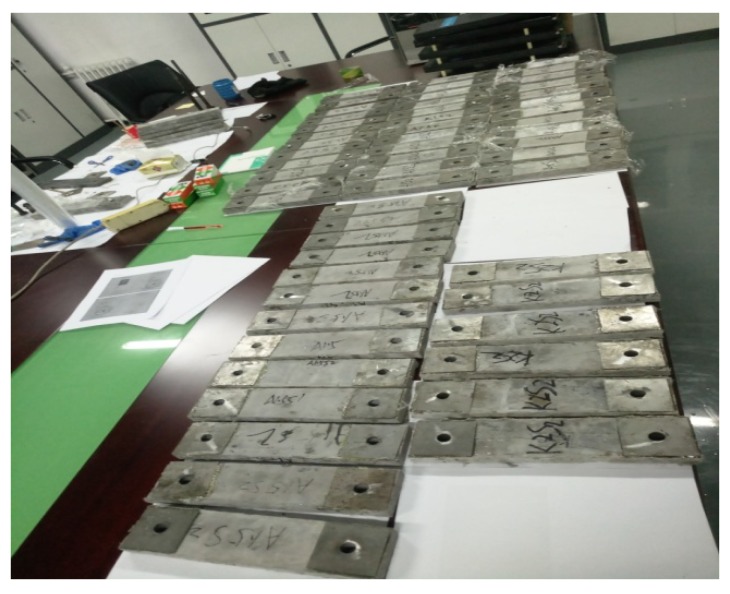
Specimens with the end reinforcement.

**Figure 4 materials-12-02666-f004:**
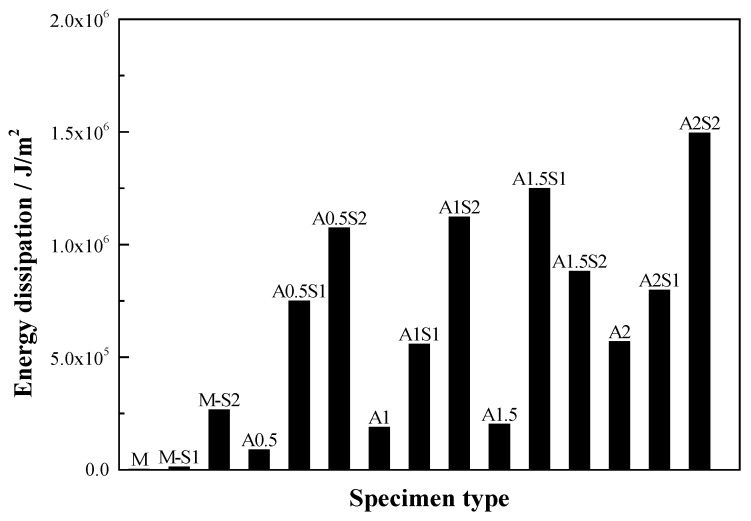
Comparison of energy dissipation of different types of steel grid-PVA fiber reinforced ECC specimens.

**Figure 5 materials-12-02666-f005:**
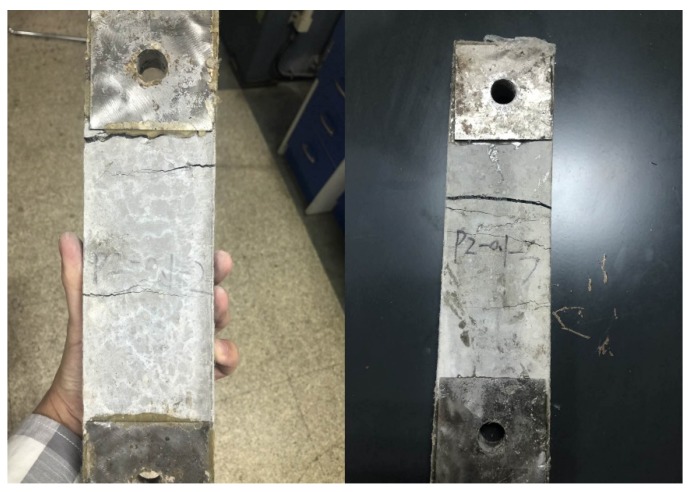
Post-failure specimens of steel grid-PVA fiber reinforced ECC with multiple cracks.

**Figure 6 materials-12-02666-f006:**
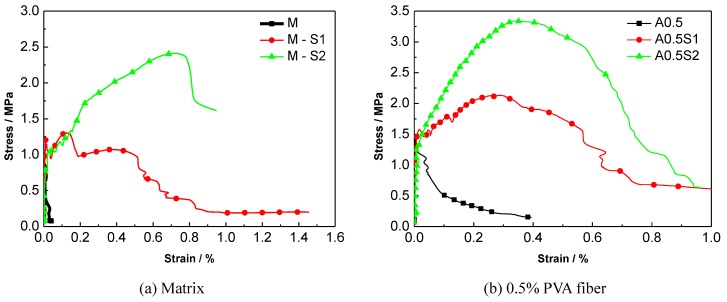
Comparison of stress–strain curves of ECC specimens with different numbers of steel-grid layers. (**a**) Matrix; (**b**) 0.5% PVA fiber; (**c**) 1% PVA fiber; (**d**) 1.5% PVA fiber; (**e**) 2% PVA fiber.

**Figure 7 materials-12-02666-f007:**
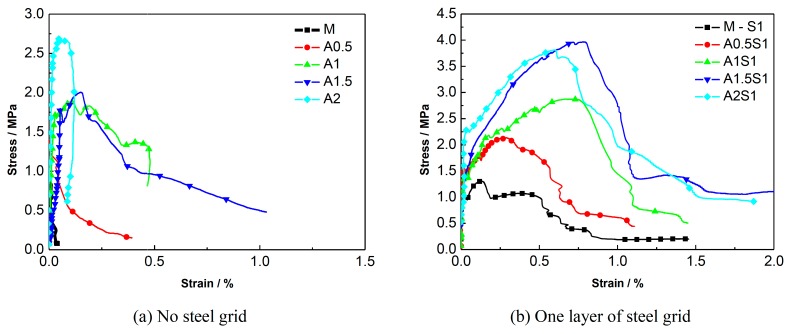
Comparison of stress-strain curves of ECC specimens with different PVA fiber volume fractions. (**a**) No steel grid; (**b**) one layer of steel grid; (**c**) two layers of steel grid.

**Figure 8 materials-12-02666-f008:**
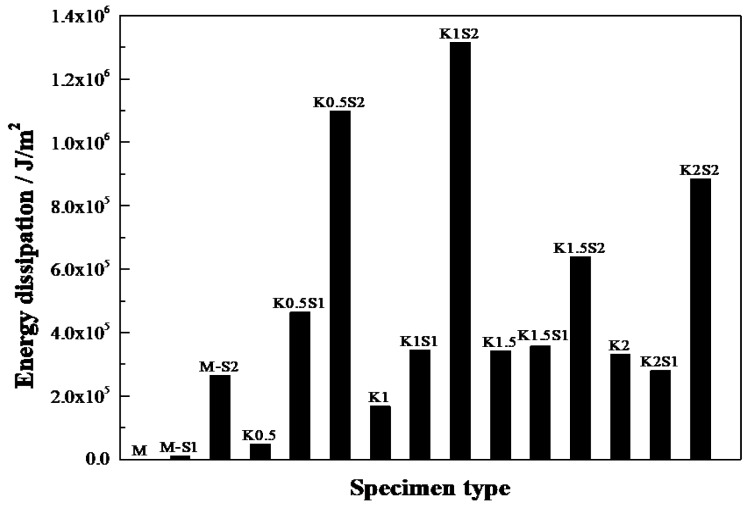
Comparison of energy dissipation of different types of steel grid-KEVLAR fiber reinforced ECC specimens.

**Figure 9 materials-12-02666-f009:**
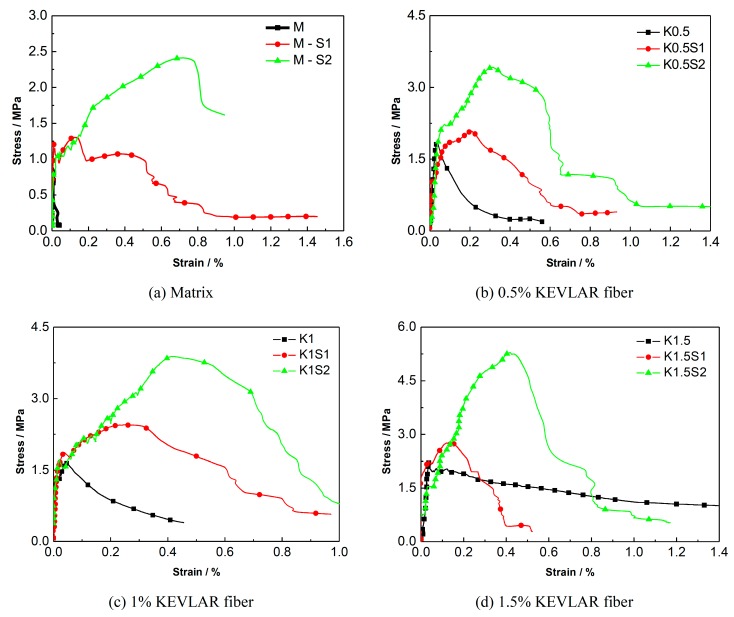
Comparison of stress-strain curves of ECC specimens with different numbers of steel-grid layers. (**a**) Matrix; (**b**) 0.5% KEVLAR fiber; (**c**) 1% KEVLAR fiber; (**d**) 1.5% KEVLAR fiber; (**e**) 2% KEVLAR fiber.

**Figure 10 materials-12-02666-f010:**
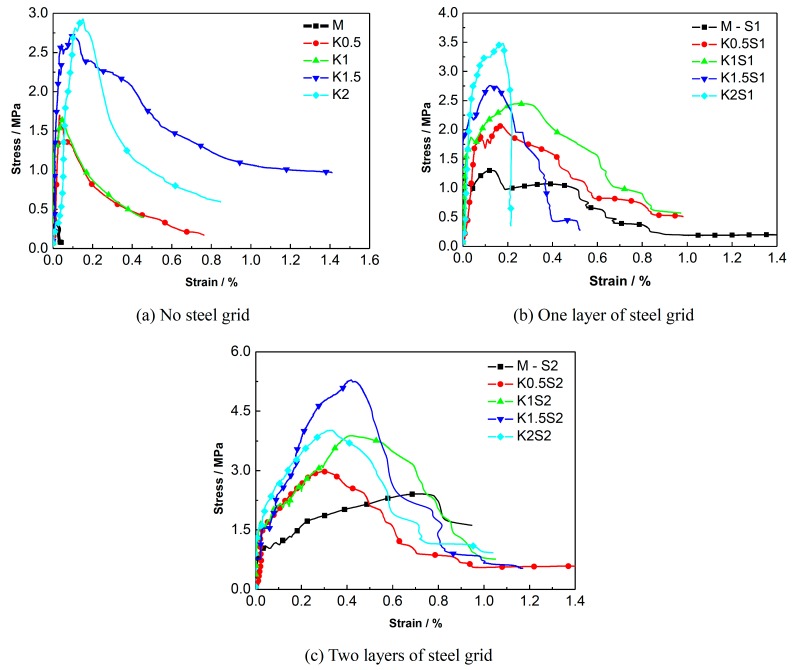
Comparison of stress-strain curves of ECC specimens with different KEVLAR fiber volume fractions. (**a**) No steel grid; (**b**) One layer of steel grid; (**c**) Two layers of steel grid.

**Figure 11 materials-12-02666-f011:**
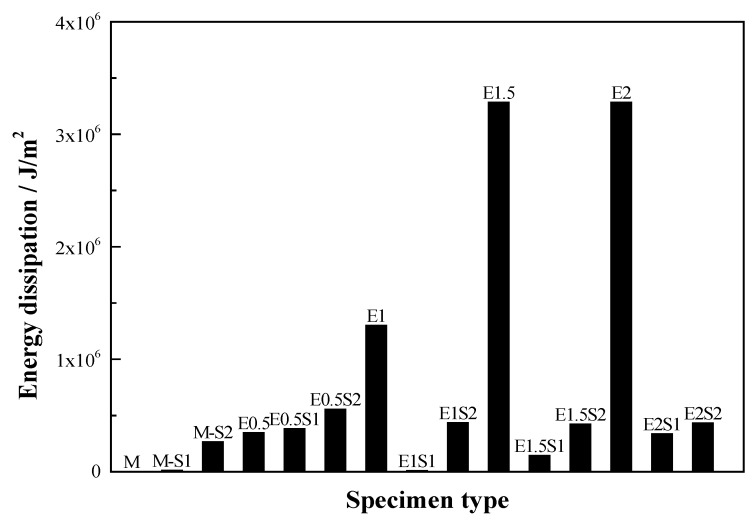
Comparison of energy dissipation of different types of steel grid-PE fiber reinforced ECC specimens.

**Figure 12 materials-12-02666-f012:**
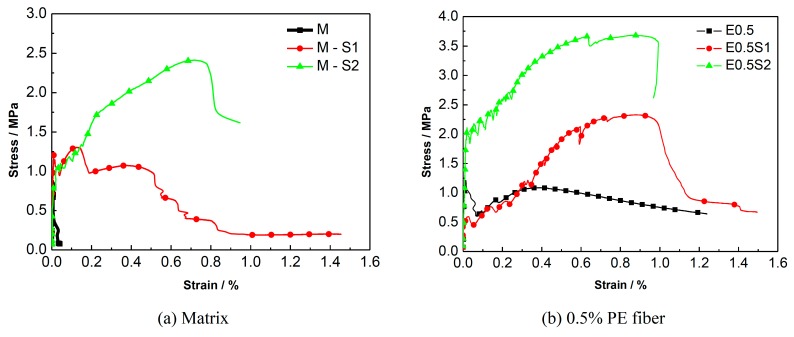
Comparison of stress-strain curves of ECC specimens with different numbers of steel-grid layers. (**a**) Matrix; (**b**) 0.5% PE fiber; (**c**) 1% PE fiber; (**d**) 1.5% PE fiber; (**e**) 2% PE fiber.

**Figure 13 materials-12-02666-f013:**
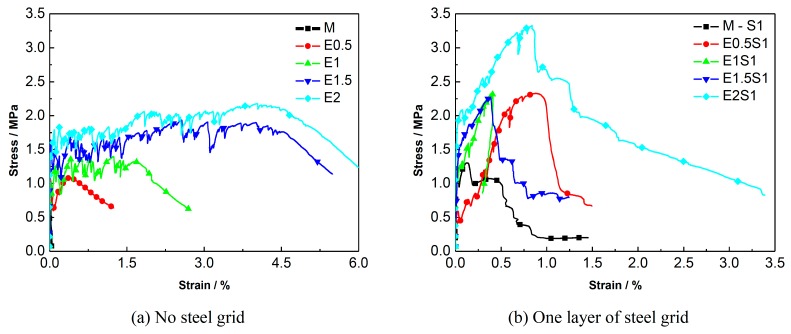
Comparison of stress-strain curves of ECC specimens with different PE fiber volume fractions. (**a**) No steel grid; (**b**) One layer of steel grid; (**c**) Two layers of steel grid.

**Figure 14 materials-12-02666-f014:**
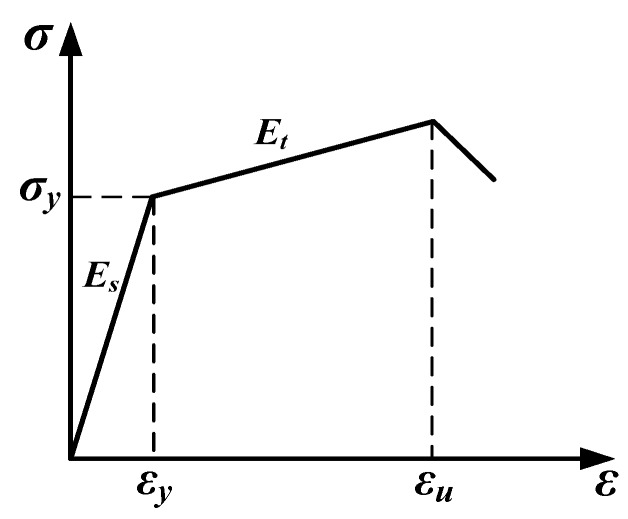
Stress-strain relation of steel grid for the tensile loading.

**Table 1 materials-12-02666-t001:** Physical and mechanical properties of P.O42.5 ordinary Portland cement.

Specific Surface Area	Initial Setting Time	Final Setting Time	Compression Strength (3 Days)	Bending Strength (3 Days)
(m^2^/kg)	(min)	(min)	(MPa)	(MPa)
381	181	243	23.5	5.3

**Table 2 materials-12-02666-t002:** Basic physical parameters of PVA fiber.

Diameter	Standard Length	Tensile Strength	Elongation Ratio	Elastic Modulus	Density
(μm)	(mm)	(MPa)		(GPa)	(g/cm^3^)
40	12	1560	6.5%	41	1.3

**Table 3 materials-12-02666-t003:** Basic physical parameters of KEVLAR fiber.

Tensile Strength	Elastic Modulus	Elongation Ratio	Standard Length	Density
(MPa)	(GPa)		(mm)	(g/cm^3^)
2920	70.5	3.6%	12	1.44

**Table 4 materials-12-02666-t004:** Basic physical parameters of PE fiber.

Tensile Strength	Elastic Modulus	Elongation Ratio	Standard Length	Density
(GPa)	(GPa)		(mm)	(g/cm^3^)
2.18	66	3.5%	12	0.97

**Table 5 materials-12-02666-t005:** Mass mixture proportions for the matrix of ECC.

Cement	Siliceous Fly Ash	Water	Superplasticizer
1.0	0.11	0.3	0.013

**Table 6 materials-12-02666-t006:** Quasi-static tensile test results of the steel grid-PVA fiber reinforced ECC.

Specimen Type	Initial Cracking Stress (MPa)	Peak Stress (MPa)	Ultimate Strain (%)	Energy Dissipation (J/m^2^)
M	0.98	0.98	0.01	1578
M-S1	0.86	1.29	0.72	11,896
M-S2	1.77	2.33	0.32	265,793
A0.5	1.39	1.39	0.05	88,070
A0.5S1	1.56	2.13	0.58	749,070
A0.5S2	3.34	3.34	0.60	1,072,785
A1	1.77	1.92	0.35	188,565
A1S1	2.05	2.58	0.28	557,326
A1S2	1.58	2.99	0.50	1,121,904
A1.5	1.78	2.00	0.23	201,600
A1.5S1	3.89	3.90	1.3	1,248,030
A1.5S2	5.14	5.14	0.64	880,790
A2	2.63	2.80	0.39	568,008
A2S1	3.93	3.93	0.67	796,900
A2S2	5.73	5.73	1.08	1,494,499

**Table 7 materials-12-02666-t007:** Quasi-static tensile test results of the steel grid-KEVLAR fiber reinforced ECC.

Specimen Type	Initial Cracking Stress (MPa)	Peak Stress (MPa)	Ultimate Strain (%)	Energy Dissipation (J/m^2^)
M	0.98	0.98	0.01	1578
M-S1	0.86	1.29	0.72	11,896
M-S2	1.77	2.33	0.32	265,793
K0.5	1.79	1.79	0.06	49,591
K0.5S1	1.93	2.07	0.16	464,231
K0.5S2	3.28	3.28	0.55	1,099,560
K1	1.6	1.6	0.12	168,336
K1S1	1.88	2.45	0.42	345,744
K1S2	3.81	3.81	0.59	1,314,334
K1.5	2.21	2.21	0.26	343,120
K1.5S1	2.25	2.76	0.22	357,616
K1.5S2	4.48	4.88	0.45	639,744
K2	2.93	2.93	0.24	332,010
K2S1	3.05	3.39	0.35	280,896
K2S2	4.15	4.15	0.29	884,450

**Table 8 materials-12-02666-t008:** Quasi-static tensile test results of the steel grid-PE fiber reinforced ECC.

Specimen Type	Initial Cracking Stress (MPa)	Peak Stress (MPa)	Ultimate Strain (%)	Energy Dissipation (J/m^2^)
M	0.98	0.98	0.01	1578
M-S1	0.86	1.29	0.72	11,896
M-S2	1.77	2.33	0.32	265,793
E0.5	1.21	1.21	0.24	347,384
E0.5S1	1.35	2.19	1.15	381,290
E0.5S2	1.52	3.59	0.50	556,080
E1	1.01	1.40	1.98	1,300,963
E1S1	1.17	2.32	0.4	8278.26
E1S2	3.60	3.60	0.51	436,100
E1.5	1.64	1.94	4.87	3,284,074
E1.5S1	2.67	2.97	0.58	142,450
E1.5S2	3.96	3.96	0.58	422,240
E2	1.8	2.18	5.18	3,284,074
E2S1	3.24	3.33	1.02	335,440
E2S2	4.34	4.34	0.79	432,075

**Table 9 materials-12-02666-t009:** Comparison of analytical and experimental initial tensile strengths of steel grid-PE fiber reinforced ECC.

Specimen Type	Analytical Initial Tensile Strength (MPa)	Experimental Initial Tensile Strength(MPa)	Error (%)
E0.5S1	1.18	1.35	−12.6
E0.5S2	1.60	1.52	5.3
E1S1	2.09	1.17	78.6
E1S2	2.77	3.60	−23.1
E1.5S1	2.77	2.67	3.7
E1.5S2	3.44	3.96	−13.1
E2S1	3.43	3.24	5.9
E2S2	4.11	4.34	−5.3

**Table 10 materials-12-02666-t010:** Comparison of analytical and experimental ultimate tensile strengths of steel grid-PE fiber reinforced ECC.

Specimen Type	Analytical Ultimate Tensile Strength (MPa)	Experimental Ultimate Tensile Strength (MPa)	Error (%)
E0.5S1	2.86	2.19	30.6
E0.5S2	4.73	3.59	31.8
E1S1	3.84	2.32	65.5
E1S2	5.71	3.60	58.6
E1.5S1	4.83	2.97	62.6
E1.5S2	6.70	3.96	69.2
E2S1	5.81	3.33	74.5
E2S2	7.68	4.34	77.0

Note: Error = (analytical results−experimental results)/experimental results × 100%.

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
