# Peer review of "Experimental Investigation on the Quasi-Static Tensile Capacity of Engineered Cementitious Composites Reinforced with Steel Grid and Fibers"

_materials, 2019, doi:10.3390/ma12172666_

Round 1

Reviewer 1 Report

In this paper, the authors summarize the results from a large series of tensile-testing experiments on engineered cementitious composites (ECCs) containing different types and amounts of fibers as well as different numbers of steel-grid layers. ECC is of significant interest to the civil engineering community and experimental results comparing the performance of mixture variations will be helpful for the research community. The paper contained a significant amount of English language errors, which I have marked (including suggested revisions) in my review (attached as a PDF with tracked changes). The attached PDF document also contains my detailed technical comments/criticisms, of which I will only mention a few below.

Given the large amount of experimental data contained in this paper and the importance of the studied material, the paper has great potential. From my perspective, however, the most interesting questions remain unanswered. For instance, the data shows that the addition of a steel grid to the samples can sometimes reduce ductility, which is counter-intuitive. I did not feel that a sufficiently detailed discussion of why this might occur was included in the paper. I also felt that the paper lacked clear “take-aways”, such as recommendations about under what circumstances one type of fibers should be used instead of another or when steel grids should be included in construction of ECC components and when they should not.

Thus, in its present state, I would not recommend immediate publication of the paper. This has less to do with the quality of the results and more to do with the fact that I believe the significant potential of this paper has not yet been fully realized.

Author Response

Dear Sir\Madam:

Thank you very much for reviewing the manuscript “Experimental Investigation on Static Tensile Capability of Engineered Cementitious Composites Reinforced with Steel Grid and Fibers”, which was submitted to the Materials and giving important comments which are very beneficial to the paper improvement. All the comments are well accepted and addressed in the revised manuscript. The revisions to English language errors are also made according to the suggestions of the reviewer. The revisions are highlighted with red in the revised manuscript. Followed please find the detailed response to each comment.

(1) Throughout the paper the term “capability” is used. “Capability” is not typically used for engineering applications. I think that the meaning that the authors are trying to convey is better expressed by the term “capacity” or by the term "strength".

Ans: Accepted.

The term “capacity” is used instead of the term “capability” in the revised manuscript.

(2) The authors refer to “Static” tensile capacities and “Static” loading. Actually, it is more-or-less impossible to actually measure true “Static” capacity, since that implies constant, unchanging load, generally over years. For the kind of experiments discussed here, I recommend using the term “quasi-static” instead of “static”, which is the typical terminology for such scenarios.

Ans: Accepted.

The term “quasi-static” is used instead of the term “static” in the revised manuscript.

(3) I do not know what the usual fiber volume fraction is, so I do not know whether 2% is a large increase or not. Remove increase or state the usual fiber volume fraction.

Ans: Accepted.

The term “increase” refers to the change of the fiber volume fraction, such as from 0.5% to 1%, or from 1% to 2%. When the fiber volume fraction increases, the reinforced ECC will acquire a higher ultimate tensile strength.

(4) Please include a citation for this first sentence.

Ans: Accepted.

The reference [1] can serve as the support of the first sentence.

(5) The authors refer to both “polymer” and “polyethylene” fibers in a list. I consider polyethylene to be a type of polymer and would, thus, recommend that only “polymer” be included in the list.

Ans: Accepted.

The term “polyethylene” has been deleted and only “polymer” is included in the list. Please see line 39 of the revised manuscript.

(6) What are “hybrid fibers”? Please explain in the text of the paper.

Ans: Accepted.

The term “hybrid fibers” means the combination of different types of fibers. This has been expressed in the revised manuscript (line 41).

(7) Throughout the paper the authors refer sometimes to “ECC”, sometimes to “engineered cementitious composites”, and sometimes to “Cementitious Composites”. Please use the term “ECC” throughout for consistency.

Ans: Accepted.

The term “ECC” is used throughout the revised manuscript.

(8) "are not sufficient to support it to serve as an individual structural material" seems unclear/incorrect. Normal concrete is used as a structural material, even though it has far lower strength than ECC. Perhaps the authors mean that the tensile strength of ECC is not high enough to allow it to carry significant structural tensile loads.

Ans: Accepted.

We mean that the tensile strength and ductility of ECC is not high enough to allow it to resist the strong blast and impact loads. This has been expressed in the revised manuscript (line 73-74).

(9) I find the notation for the mixture proportion of cement, silica ash, water, and superplasticizer “1:0.11:0.013:0.3.” difficult to read. I recommend placing this information in a table rather than in the text of a sentence.

Ans: Accepted.

The mass mixture proportion of cement, silica ash, water and superplasticizer for the matrix of the ECC has been listed in Table 5.

(10) The authors refer to “concreting time”. I think that the term “curing time” is much more common and easily understandable. I am also unfamiliar with the technical distinction between "initial curing time" and "ultimate curing time". Please describe this in the text or provide a reference that that does.

Ans: Accepted.

The “initial concreting time” and “ultimate concreting time” are expressed as “initial setting time” and “final setting time”, respectively. The technical meanings of the two terms have also been described in the revised manuscript (line 93-95).

(11) Tables 2-4. The authors refer to “Young’s modulus”, “Tensile modulus”, and “Elastic modulus”. Are these all the same property? If so, then please use consistent notation.

Ans: Accepted.

The consistent notations, including “Tensile strength” and “Elastic modulus” have been used in Tables 2-4.

(12) Tables 3-4. No information is given about the size, diameter, or length of the KEVLAR and PE fibers. Please include the information if it is available.

Ans: Accepted.

The information about the standard length of the KEVLAR and PE fibers has been presented in Tables 3-4.

(13) Table 4: “Breaking strength” is included in addition to “Ultimate tensile strength”. I do not think that this is necessary and recommend removing “breaking strength” since the ultimate tensile strength is directly calculated from the breaking strength.

Ans: Accepted.

The information about the breaking strength has been removed from Table 4.

(14) I do not see any wire grid in the center of the mold shown in Fig. 3. Are the specimens on the left and right of the mold sawn out of a larger piece? If so, this sawing procedure should also be mentioned in the specimen fabrication description.

Ans: Accepted.

The specimens with one layer and two layers of steel-grid are not sawn out of a larger piece. They are casted in the mold separately.

(15) Is this a standard procedure for defining ultimate strain? If so, then please provide a citation. If not, then please explain why you chose this method for defining ultimate strain.

Ans: Accepted.

The explanation for the definition of ultimate strain has been presented in the revised manuscript (line 150-152).

(16) All types of specimens split completely at the ultimate failure state. Thus, this sentence has no significant meaning. Remove or rephrase to express the intended meaning more clearly.

Ans: Accepted.

This sentence has been removed from the revised manuscript.

(17) It appears to me that the ultimate strain increases by a few hundred percent. I think that the numbers you mention here are the increase in absolute strain percentage.

For the peak tensile stress results, you quote the increases in terms of percentage of the matrix value. I recommend that you do the same for the strain (i.e. ((ultimate steel grid sample strain)/(ultimate matrix strain))*100). If you would rather use the absolute strain values instead, you should explicitly state that the ultimate strain of the steel-grid specimens is "increased by 0.6% and 0.8% absolute strain compared to the matrix specimens".

Ans: Accepted.

The description using the absolute strain values advised by the reviewer is adopted in the revised manuscript (line 173-175).

(18) You have just shown that the steel grids have an enormous impact on the ultimate strain, but now you say that the ultimate strain depends primarily on the matrix. This appears to be a contradiction. Please reformulate in order to better express your intended meaning.

Ans: Accepted.

The description contradict with the previous conclusions is removed from the revised manuscript.

(19) Please quantify. How much poorer is the KEVLAR performance (for instance in %).

Ans: Accepted.

The quantitative comparison of the ultimate strains of KEVLAR-ECC and PVA-ECC has been presented in the revised manuscript (line 211-212).

(20) This is not clear to me. I seem to see strain hardening behavior for all added ECC fiber percentages (for instance Fig. 14b). Please remove statement or explain in more detail.

Ans: Accepted.

The statement has been removed from the revised manuscript.

(21) High compared to what. High is always relative. Please revise (for instance, higher strength and ductility than the ECC matrix).

Ans: Accepted.

The statement has been revised by a comparative way in the revised manuscript (line 355-356).

(22) I disagree with this statement. The steel grid reduced the ductility of the ECC with PE fibers even though there was a significant tensile strength increase. Please modify.

Ans: Accepted.

The statement has been revised in the revised manuscript (line 356-357). Only the tensile capacity of the steel grid-PE fiber reinforced ECC is described.

(23) This cannot be observed by the reader since there are no pictures of failed specimens. I think that some pictures of post-failure specimens would be useful in order to see the distributed cracking. The authors also mention that there is a debonding of the steel grid from the concrete matrix in the results section of the paper. Post failure images of the specimens would presumably allow the reader to better understand what this debonding practically looks like.

Ans: Accepted.

Some pictures of post-failure specimens with multiple cracks of the steel grid-PVA fiber reinforced ECC have been presented in the revised manuscript (Fig.5).

(24) From my perspective, the most interesting questions remain unanswered.

1) Why does the steel grid reduce ductility for the PE fiber ECC? This is counter to my expectations. I did not find the explanation in the results section to be comprehensive. I think that further discussion of the phenomena that cause this phenomena (probably including photographs) would be useful.

2) What is the "take-away" from this paper? Under what circumstances is it better to use PVA fibers instead of KEVLAR or PE fibers, for instance? Are there ever cases in which only one steel grid layer should be used, or should two layers always be used?

Ans: Accepted.

1) The test results indicate that the steel grid reduce ductility for the PE fiber ECC for some loading cases. For the specimens with a fiber volume fraction of 1-2%, the ultimate strain decreases when the steel grid is added to the matrix. After the peak stress, the steel grid will separate from the matrix, and this leads to the deformation inconsistency between the steel grid and the matrix. Under this circumstance the ultimate strain mainly depends on the deformation capacity of the steel grid. But the deformation capacity of the steel grid is weaker than that of the PE-ECC with a relatively high fiber volume fraction, such as 1.5% or 2%. This has also been expressed in the revised manuscript (line 234-238).

2) Some engineering suggestions have been presented in the revised manuscript (line 363-366, 373-376). The phenomenon of multiple-cracking can be observed for steel grid-PVA fiber reinforced ECC. Under those circumstances when excellent ductility and energy dissipation performance are required, it is better to use PVA fibers instead of KEVLAR or PE fibers. A more remarkable increase of ultimate tensile strength can be obtained by adding two layers of steel grid than one layer of steel grid, but it is more difficult to have a firm bonding with the ECC matrix for two layers of steel grid than one layer of steel grid.

Reviewer 2 Report

The paper deals with the modern type of construction composites - engineered cementitious composites (ECC). This group of materials is intensively developed recently, thus the topic of the paper is interesting and up-to-date. Not all of the findings and assumptions of the work are really original (e.g., two layers of reinforcement are better than one layer, etc.), nonetheless the text is generally well prepared, the experiments are carefully planned and well described, and at least part of the conclusions are important from the practical point of view. Taking above into consideration, I recommend to publish the paper after the following corrections:

- "polymer fiber" and "polyethylene fiber" should not be distinguished, as polyethylene is also a polymer;

- the statement that "ECC are not sufficient (...) to serve as an individual structural material" needs explanation - there are known ECC which can play the structural function;

- what do Authors mean by "silica ash"? Silica fume or siliceous fly ash?

- "concreting time" is a technological term, which does not refer to the cement, only to the concrete mix, and depends mainly on your decision; the Authors mean "setting time";

- last, I really do not think that in the scientific paper should be presented the images of the moulds or testing machine, so I suggest to remove at least the Figs. 3, 5, 6.

Author Response

Dear Sir\Madam:

Thank you very much for reviewing the manuscript “Experimental Investigation on Static Tensile Capability of Engineered Cementitious Composites Reinforced with Steel Grid and Fibers”, which was submitted to the Materials and giving important comments which are very beneficial to the paper improvement. All the comments are well accepted and addressed in the revised manuscript. The revisions are highlighted with red in the revised manuscript. Followed please find the detailed response to each comment.

(1) "polymer fiber" and "polyethylene fiber" should not be distinguished, as polyethylene is also a polymer.

Ans: Accepted.

The term “polyethylene fiber” has been deleted and only “polymer fiber” is included in the text. Please see line 39 of the revised manuscript.

(2) the statement that "ECC are not sufficient (...) to serve as an individual structural material" needs explanation - there are known ECC which can play the structural function.

Ans: Accepted.

We mean that the tensile strength and ductility of ECC is not high enough to allow it to resist the strong blast and impact loads. This has been expressed in the revised manuscript (line 73-74).

(3) what do Authors mean by "silica ash"? Silica fume or siliceous fly ash?

Ans: Accepted.

We mean the siliceous fly ash by “silica ash”.

(4) "concreting time" is a technological term, which does not refer to the cement, only to the concrete mix, and depends mainly on your decision; the Authors mean "setting time".

Ans: Accepted.

The “initial concreting time” and “ultimate concreting time” are expressed as “initial setting time” and “final setting time”, respectively.

(5) last, I really do not think that in the scientific paper should be presented the images of the molds or testing machine, so I suggest to remove at least the Figs. 3, 5, 6.

Ans: Accepted.

Figs.3, 5, 6 have been removed from the revised manuscript. The figures throughout the manuscript are renumbered.

Round 2

Reviewer 1 Report

This paper has been significantly improved since the first round of review. A large proportion of the text has been revised and several pages have been added, including new paper sections about the proposed mechanical prediction equations. These changes make the paper much more innovative and useful, adding information about interpreting the results and using them to validate the mechanical equations. However, within this revised paper a significant number of errors and unresolved questions remain. The attached PDF document contains my detailed technical comments/criticisms. I believe that some of these questions (such as those regarding derivation of the equations) are very important to resolve and I would not recommend immediate publication of the paper until they have been addressed.

Author Response

Dear Sir\Madam:

Thank you very much for reviewing the manuscript “Experimental Investigation on Quasi-Static Tensile Capacity of Engineered Cementitious Composites Reinforced with Steel Grid and Fibers”, which was submitted to the Materials and giving important comments which are very beneficial to the paper improvement. All the comments are well accepted and addressed in the revised manuscript. The revisions are highlighted with red in the revised manuscript. Followed please find the detailed response to each comment.

(1) I do not know what is meant here. Do you mean "For all of the tested fiber types, a volume fraction between 1.5% and 2% made the reinforced ECC gain the best tensile strength"?.

Ans: Accepted.

We mean that for all of the tested fiber types, PVA fiber, KEVLAR fiber and PE fiber, a volume fraction between 1.5% and 2% made the reinforced ECC gain the best tensile strength. This conclusion is based on the test results of the current study. In the current test study, the maximal fiber volume fraction is 2%.

(2) Adobe is having trouble marking the text in this sentence so I will recommend changes using a comment. I recommend instead writing:

"and the term "final setting time" refers to the time required for the cement slurry to completely lose its plasticity and to begin to exhibit considerable strength.".

Ans: Accepted.

The expression recommended by the reviewer has been adopted in the revised manuscript (line 94-95).

(3) I would not really call this "strain-hardening". Strain hardening cannot happen after peak stress. The term "hardening" means that there is an increase in material stress during plastic deformation. I do think, however, that I understand what you are trying to say and recommend something like the following:

"Because ECC exhibits considerable strength and ductility even after reaching its peak tensile stress, during loading the failure of ECC is considered to occur when the tensile stress has descended to 80% of its peak value.".

Ans: Accepted.

The expression recommended by the reviewer has been adopted in the revised manuscript (line 150-152).

(4) My understanding is that the "matrix" test results come from only one specimen.

Ans: Accepted.

In the current study, the “matrix” test results come from the average value of the tension test results of a group of matrix specimens.

(5) It looks to me like the increase relative to the pure matrix specimen is much larger than 2 or 3.5 times based on Fig.9a.

Ans: Accepted.

This result is applicable for the KEVLAR-ECC. For the KEVLAR-ECC specimens, the ultimate strain increases by about 2 or 3.5 times compared to the corresponding matrix specimen by adding one layer or two layers of steel grid, respectively. This has been expressed in the revised manuscript (line 206-207). For the pure matrix specimens, the ultimate strain is too small to present a comparison with other specimens.

(6) My assumption is that all of these "increases by about X%" comparisons throughout the paper are relative to the unreinforced matrix specimen (hence my recommended text additions). If they are calculated in some other way (for instance an average increase calculated from samples with 0, 1, and 2 grid layers), then you should explicitly state how the comparison numbers are calculated in the text.

Ans: Accepted.

The "increases by about X%" comparisons throughout the paper are relative to the matrix specimen without fibers addition. This has been added to the revised manuscript (line 180, 211, 247).

(7) Please state where Eq.4 comes from. Is it from the literature (if so, please provide citation)? Is it from a set of experiments (if so, please describe)?

Ans: Accepted.

A figure is presented in the revised manuscript to illustrate the stress-strain relation of the steel grid for the tensile loading. Please see Fig.14 in the revised manuscript.

(8) I do not understand the "f" in the term e^(f*phi). Should the "f" be a subscript instead of a superscript? I do not see any definition for the term "f" in the text.

Ans: Accepted.

Eqs. (8) and (9) are revised and the wrong terms in the equations have been removed.

(9) I have a couple of comments about this equation.

I do not understand why is the limit of Eq. 13 (delta*) is not the same as Eq. 8 (delta0)?

I do not understand the appearance of g here. When you derive a new equation from three previous equations, there should not be any additional terms in the new equation that were not in the original equations. What is g, where did it come from, and how do you calculate it?

I am admittedly not a mathematician. I tried to re-derive Eq.13 from Eqs 8, 11, and 12. I was not able to obtain the equation in the form described here with my derivation. Please provide a copy of your derivation either specifically to the reviewers, so that we can check it, or (even better) as a public appendix to the paper.

Ans: Accepted.

delta* in Eq.13 is the crack split displacement of the fiber with a length of l/2 when full debonding occurs, and delta0 in Eq.8 is the crack split displacement of the fiber with a length of l when full debonding occurs.

The expression of the buffer factor g is presented in the revised manuscript (Eq.14).

Substituting Eqs. (8), (11) and (12) into Eq. (10) and carrying out the integral operation can result in Eq.13.

(10) Why does the Ktip term disappear when c is greater than w? This was not clear to me based on the previous text.

Ans: Accepted.

When the length of the crack c is greater than the width of the specimen w, the crack end toughness is not taken into consideration, and the Ktip term is deleted from the tensile stress-crack split displacement relation. This has been expressed in the revised manuscript (line 320-322).

(11) The initial tensile strength results are not totally accurate. If they were, there would be no error. I do not think that this sentence is necessary and can be removed.

Ans: Accepted.

This sentence has been removed from the revised manuscript.

(12) I do not understand what you are trying to say with this second sentence. It appears to be written in contrast two the first sentence of (1). Based my the figures, it appears to me that the steel grid-PE fiber reinforced ECC also has higher strength and ductility than the ECC matrix. If this is true, why is it not listed in the first sentence with PVA and Kevlar and what is the purpose of the second sentence? In this context "capacity" is also a little bit vague. If you are specifically referring to strength or ductility, please state them explicitly.

Ans: Accepted.

These sentences have been rewritten in the revised manuscript (line 368-370).

(13) I still do not understand how these numbers are calculated. I think that you must be averaging the increase in tensile strengths over all fiber percentages, but I do not think that this is ever explicitly stated in the paper. Please state somewhere in the paper how this calculation in made.

Ans: Accepted.

These sentences have been rewritten. The maximal peak tensile stress increases by adding one layer or two layers of steel grid and the corresponding fiber volume fractions for different types of steel grid-fiber reinforced ECCs are expressed in the revised manuscript (line 170-172, 203-205, 232-234, 372-380).

(14) It is not clear to me why the bond would be impacted by the number of layers. Please explain.

Ans: Accepted.

When two layers of steel grid are used, the space between the steel grids and the ECC matrix is smaller than that when one layer of steel grid is used, so it is more difficult to have a firm bonding with the ECC matrix for two layers of steel grid than one layer of steel grid.

(15) This is contrary to my understanding. The PE fiber samples exhibited a significantly higher strain (around 6% - Fig 13a) and higher energy dissipation (around 3.5x10^6 J/m^2 - Fig 11) than any of the other materials. Thus, I would expect that for circumstances when excellent ductility and energy dissipation performance were required, PE fibers are better to use.

Ans: Accepted.

This has been expressed in the revised manuscript (line 390-392).

(16) Sentence unnecessary. Already sufficiently explained by the previous sentence.

Ans: Accepted.

This sentence has been removed from the revised manuscript.
